# High Heme and Low Heme Oxygenase-1 Are Associated with Mast Cell Activation/Degranulation in HIV-Induced Chronic Widespread Pain

**DOI:** 10.3390/antiox12061213

**Published:** 2023-06-03

**Authors:** Tanima Chatterjee, Itika Arora, Lilly Underwood, Anastasiia Gryshyna, Terry L. Lewis, Juan Xavier Masjoan Juncos, Burel R. Goodin, Sonya Heath, Saurabh Aggarwal

**Affiliations:** 1Department of Anesthesiology and Perioperative Medicine, Division of Molecular and Translational Biomedicine, University of Alabama at Birmingham, Birmingham, AL 35233, USA; tchatterjee@uabmc.edu (T.C.); lillyunderwood@uabmc.edu (L.U.); gryshyna@uab.edu (A.G.); terrylewis@uabmc.edu (T.L.L.); jxjuncos@uabmc.edu (J.X.M.J.); 2Division of Developmental Biology and the Reproductive Sciences Center, Cincinnati Children’s Hospital, Cincinnati, OH 45229, USA; itika.arora@cchmc.org; 3Washington University Pain Center, Department of Anesthesiology, Washington University in St. Louis, St. Louis, MO 98105, USA; burel@wustl.edu; 4Division of Infectious Disease, University of Alabama at Birmingham, Birmingham, AL 35233, USA; slheath@uabmc.edu

**Keywords:** HIV, cell-free heme, heme oxygenase, mast cells, chronic widespread pain

## Abstract

An overwhelming number of people with HIV (PWH) experience chronic widespread pain (CWP) throughout their lifetimes. Previously, we demonstrated that PWH with CWP have increased hemolysis and attenuated heme oxygenase 1 (HO-1) levels. HO-1 degrades reactive, cell-free heme into antioxidants like biliverdin and carbon monoxide (CO). We found that high heme or low HO-1 caused hyperalgesia in animals, likely through multiple mechanisms. In this study, we hypothesized that high heme or low HO-1 caused mast cell activation/degranulation, resulting in the release of pain mediators like histamine and bradykinin. PWH who self-report CWP were recruited from the University of Alabama at Birmingham HIV clinic. Animal models included HO-1^−/−^ mice and hemolytic mice, where C57BL/6 mice were injected intraperitoneally with phenylhydrazine hydrochloride (PHZ). Results demonstrated that plasma histamine and bradykinin were elevated in PWH with CWP. These pain mediators were also high in HO-1^−/−^ mice and in hemolytic mice. Both in vivo and in vitro (RBL-2H3 mast cells), heme-induced mast cell degranulation was inhibited by treatment with CORM-A1, a CO donor. CORM-A1 also attenuated mechanical and thermal (cold) allodynia in hemolytic mice. Together, the data suggest that mast cell activation secondary to high heme or low HO-1 seen in cells and animals correlates with elevated plasma levels of heme, histamine, and bradykinin in PWH with CWP.

## 1. Introduction

Chronic widespread pain (CWP) is a debilitating condition that affects many individuals but has an increased prevalence in people diagnosed with human immunodeficiency virus-1 (HIV-1) [1,2]. Although the effectiveness of anti-retroviral therapy (ART) in HIV has been clearly established, potential comorbidities associated with HIV-1 infection, such as CWP, still impair quality of life [3]. The prevalence of CWP in people with HIV (PWH) ranges between 25% and 85%; however, current pain management strategies are not sufficient to alleviate pain [2,4,5,6]. There is a heavy reliance on opioids to alleviate pain among PWH [7]. A history of opioid substance abuse in PWH requires higher doses for pain management and consequently increases the risk of opioid dependence. Therefore, the development of novel therapeutics is necessary to decrease opioid-related substance abuse in this population.

Mast cells play an important role in stress-mediated responses and are known to be activated during neuroinflammation [8,9,10]. The degranulation of mast cells causes exocytosis of membrane-bound granules, which results in the release of inflammatory mediators such as histamine, tryptase, cytokines, chemokines, leukotrienes, and prostaglandins [11,12] and the generation of bradykinin [13]. These inflammatory factors bind to nociceptors and induce neuroinflammation and pain [11]. Various stimuli can induce mast cell activation/degranulation in neuropathy. For example, in diabetic neuropathic pain, high blood glucose levels are linked to mast cell activation; in sickle cell disease (SCD), the sickling of red blood cells and resulting blood vessel occlusion activate mast cells [11,14]. 

The activation of mast cells is compounded by different pathological conditions, each with a unique molecular and cellular microenvironment [11]. IL-1 family cytokines, which are elevated in several diseases associated with chronic pain [1,15] are known activators of mast cells. Previously, we demonstrated that PWH with CWP have increased hemolysis and elevated plasma levels of cell-free heme [1]. These people also had increased plasma IL-1β levels [1]. Cell-free heme is a pro-inflammatory molecule that can induce the secretion of IL-1 family cytokines from immune cells, including mast cells, and therefore can potentially induce mast cell degranulation [12,16]. We also found that PWH with CWP have attenuated levels of the heme-degrading enzyme, heme oxygenase 1 (HO-1) [1]. HO-1 metabolizes cell-free heme into biliverdin and carbon monoxide (CO). High cell-free heme and low HO-1 were associated with hypersensitivity in animals [1,17]. 

The objective of the present study was to examine whether mast cell activation and chronic pain in HIV are associated with elevated plasma levels of cell-free heme and low expression of HO-1. In an animal model of hemolysis or HO-1 knockdown, there was an increase in mast cell activation, elevated plasma levels of histamine and bradykinin, and allodynia. We also demonstrated that the administration of the heme-scavenging enzyme, hemopexin, or the CO donor, CORM-A1, mitigated mast cell activation/degranulation and associated hypersensitivity in mice. These studies provide novel therapeutic targets, which may prove useful in mitigating pain in PWH and helping reduce opioid dependency. 

## 2. Materials and Methods

2.1 Human participants: Human subjects were classified into one of four different groups: (1) HIV-negative, pain-negative individuals. To be included in this group, these individuals did not present with chronic disease, as this may lead to increased hemolysis; (2) HIV-negative, pain-positive individuals experiencing low back pain; (3) HIV-positive, pain-negative individuals; and (4) HIV-positive, pain-positive individuals suffering from CWP. Classifications are based on work previously performed by the authors [1]. Participants positive for HIV, as well as those with HIV with CWP, were identified through the UAB Center for AIDS Research Network of Integrated Clinical Systems (CNICS) by a brief chronic pain questionnaire, which, as part of their Patient Reported Outcomes (PROs), included their history of intensity and duration of pain if present [18]. Those participants with CWP reported low back pain as the source of the majority of their pain. Therefore, in order to more closely control this group, individuals with chronic low back pain for three consecutive months or at least half of the days during the previous six months were recruited by posting flyers throughout UAB and the surrounding community. In order to be included, participants had to be free from trauma/accident and not have received any type of surgery on the lower back. 

Following the collection of blood samples from participants, RBCs were isolated from plasma by centrifugation, and the plasma was then stored at −80 °C. Repeated freeze-thaw cycles of plasma were avoided prior to the measurement of human histamine and bradykinin.

2.2 Reagents: EMEM medium (product no. 30-2003), fetal bovine serum (product no. 30-2020), dimethylsulfoxide (product no. 4-X), penicillin/streptomycin (product no. 30-2300), and trypsin (product no. 30-2101) were purchased from the American Type Culture Collection (ATCC) (Manassas, VA, USA). Hemopexin (Hx) (product no. 16-16-080513) was obtained from Athens Research and Technology (Athens, GA, USA). Phenylhydrazine hydrochloride (PHZ) (product no. 114715), CORM-A1 (product no. SML0315), and calcium ionophore A23187 (product no. C7522) were obtained from Sigma-Aldrich (St. Louis, MO, USA).

2.3 Cell culture and treatment: Rat basophilic leukemia cells (RBL-2H3) (product no. CRL-2256) were purchased from ATCC and cultured in EMEM medium supplemented with 15% FBS, 2 mM L-glutamine, 100 IU/mL penicillin, and 100 µg/mL streptomycin. Cells were seeded at a density of 5 × 10^4^ cells/well in a 96-well plate and incubated for 24 h at 37 °C with 95% humidity and 5% CO_2_. Medium was then aspirated, and cells were washed twice with Hank’s buffer with hepes (20 mM) and incubated in 100 µL growth buffer. Cells were then treated with hemin (25 µM) or vehicle (dimethyl sulfoxide, DMSO) in the presence or absence of CORM-A1 (10 mM) for 2 h. RBL-2H3 cells are known to exhibit degranulation in response to calcium ionophore but not to compound 48/80 [19]. Thus, in the following experiments, RBL-2H3 cells were stimulated with calcium ionomycin, A23187 (5 µg/mL), and used as a positive control. A23187 evokes mast cell degranulation by increasing intracellular calcium levels, either by acting as a carrier or by promoting calcium entry into cells by native calcium channels [20].

2.4 HO-1 silencing and quantification in cells: Cellular levels of HO-1 were genetically attenuated in RBL-2H3 cells using mouse heme oxygenase siRNA (r) siRNA (product no. sc-3555; Santa Cruz Biotechnology, Dallas, TX, USA) for 48 h. Control cells received scrambled siRNA (product no. sc-37007; Santa Cruz Biotechnology). Cells were homogenized in RIPA buffer containing protease inhibitors. Samples were sonicated for 3 × 10 s on ice in 1.5 mL microfuge tubes using an ultrasonic liquid processor and centrifuged at 14,000× *g* for 20 min at 4 °C. Protein concentration was measured in cleared supernatants using the Bradford assay kit (product no. 5000205, Bio-Rad Laboratories, Hercules, CA, USA). Equal amounts of protein (25 µg) were loaded in 10% Tris·HCl Criterion precast gels (product no. 567–1093, Bio-Rad Laboratories, Hercules, CA, USA) and transferred to polyvinylidene difluoride membranes (product no. 162–0177, Bio-Rad Laboratories) and immuno-stained with an anti-HO-1 antibody (1:1000; product no. ADI-SPA-896-F, Enzo Life Sciences, Farmingdale, NY, USA). Bands were detected using the ChemiDoc MP imaging system (Bio-rad) and analyzed with Image Lab Software (Bio-rad). Protein loading was normalized by re-probing the membranes with an antibody specific to β-actin.

2.5 Mouse model and treatment: Adult male and female C57BL/6 mice (20–25 g) were purchased from Jackson Labs, Farmington, CT. Dr. Anupam Agarwal of UAB supplied heme oxygenase-1 (HO-1^−/−^) mice based on a mixed background of C57BL/6xFVB along with wildtype (WT) littermates, as discussed in a previous publication [21]. Mice were given a standard diet and water ad libitum while being housed following a 12-hour light/dark cycle. An intraperitoneal injection of ketamine/xylazine was administered at the end of experimentation to reduce stress and pain during euthanasia. Mice were administered PHZ (50 mg/kg), which leads to increased hemolysis, or saline (vehicle) via intraperitoneal (IP) injection for two consecutive days. One hour following the second and final PHZ dose, mice were given hemopexin (4.0 mg/kg) or CORM-A1 (2.0 mg/kg) in 1x phosphate buffered saline (PBS) by intraperitoneal injection. The CORM-A1 injection was repeated for an additional 3 days following the initial dose. Rodent blood and tissue samples were collected 4 days after the second PHZ injection.

2.6 Heme quantification: Plasma heme concentrations in human samples and mice were measured by the QuantiChrom heme assay kit (product no. DIHM-250; BioAssay Systems, Hayward, CA, USA), according to the manufacturer’s instructions. 

2.7 Histamine, bradykinin, and β-hexosaminidase quantification: Histamine levels were measured in human and mouse plasma and in vitro from RBL-2H3 supernatant using a histamine enzyme-linked immunosorbent assay kit (product no. ENZ-KIT140A-0001, Enzo Life Sciences Inc., Farmingdale, NJ, USA) according to the manufacturer’s protocol. Bradykinin levels were measured in human and rodent plasma using a bradykinin ELISA kit (product no. ADI-900-206, Enzo Life Sciences Inc., Farmingdale, NJ, USA) according to the manufacturer’s protocol. β-hexosaminidase activity, an index of mast cell degranulation, was measured in RBL-2H3 cells treated with DMSO, hemin (25 µM), CORM-A1 (10 mM), or A23187 (5 µg/mL) for 2 h, as mentioned earlier. The reaction was terminated by incubating cells at 4 °C for 10 min. β-hexosaminidase activity was measured in cell supernatant with a β-hexosaminidase activity assay kit (product no. MET-5095, Cell Biolabs, San Diego, CA, USA) according to the manufacturer’s instructions. 

2.8 Toluidine blue staining: In vivo, rodent skin tissue was fixed in 5% buffered formalin, and sections were stained with hematoxylin and eosin dyes (H&E) and Toluidine blue. Imaging was performed using a confocal microscope (Leica, DM-5000-B). Quantification was performed by a blind operator by taking an average of mast cells in five fields per sample. Mast cells were only counted when present in the dermal layer, while mast cells present in the hypodermis were excluded from this count. The obtained values were analyzed and plotted. In vitro, RBL-2H3 cells (2 × 10^5^ cells/mL) were grown to 50% confluence on polylysine-coated glass coverslips, washed with plain EMEM, and incubated with 1% toluidine blue for 20 min at 37 °C. Cells were then fixed with methanol at −20 °C for 10 min, rinsed with PBS, and mounted for viewing in a glycerol-based medium using a confocal microscope. The normal spindle-shaped mast cells became round or tadpole-shaped cells, and the cytoplasm disappeared upon degranulation.

2.9 Behavioral study: Three different types of evoked behavioral assays were performed on mice to observe their sensitivity to mechanical, heat, and cold stimuli. Prior to conducting these test sessions, mice were placed in an enclosure (Bioseb, product no. BIO-VF-M, Pinellas Park, FL, USA) to acclimate for one hour. The enclosure was composed of a plexiglass chamber over a mesh floor, which provided easy visualization of responses and access to the paws. 

Von Frey filament test: Mechanical allodynia was assessed using the simplified up-down method [22]. In this method, the paw withdrawal threshold estimate was determined using calibrated monofilaments (product no. BIO-PVF; Aesthesio, Bioseb, Pinellas Park, USA). The test always began with monofilament 3.22. If the subject had no withdrawal response from the filament, the next highest filament size was applied. If a withdrawal response was observed, the next lowest filament size was applied. A total of five responses were recorded per hind paw. The final withdrawal score was calculated by taking an average of the final responses from both paws. Depending on responses, the used filaments could range from 1.65 to 4.31. The filament values were converted to force (grams). The data are shown as a percent change from the pretreatment baseline.

Acetone evaporation test: Thermal (cold) allodynia was tested by observing the number of responses following the application of acetone (kept at room temperature) to the center of each hind paw. Acetone was applied with a small syringe, and mice were observed for 60 s following application. A number of reactions (flicking or licking of the tested paw) were recorded. The score was 0 for no reactions, 1 for a single reaction, and 2 for two or more reactions. The test was conducted separately for each paw, and the average score for both paws was calculated. Response to the application of water (kept at room temperature) was used as a control prior to the acetone test. 

Hot plate test: Thermal (heat) hyperalgesia was determined by recording the latency to response (in seconds) when mice were placed on a plexiglass-enclosed hot plate (MazeEngineers, Skokie, IL, USA). The hot plate temperature was set to 55 °C, and individual testing began once the temperature reached that level. Each mouse was timed from their placement until they exhibited a response, which included licking/flicking of the hind paws or attempting to jump off the plate. If a mouse did not respond within 30 s, the test ended, and they were removed from the enclosure.

2.10 Statistics: Statistical analysis was performed using GraphPad Prism version 9.5 for Windows (GraphPad Software, San Diego, CA, USA). Results were expressed as the mean ± standard error of the mean (SEM). Statistical significance was determined by an unpaired *t*-test for two groups or a one-way ANOVA followed by Tukey’s post-hoc test for more than two groups; *p* < 0.05 was considered significant.

## 3. Results

3.1 People with HIV who self-report CWP have increased plasma levels of histamine and bradykinin. The role of mast cells in HIV-associated CWP is unknown. We recruited four groups of participants for our study: (1) HIV-negative individuals without pain; (2) HIV-negative individuals with chronic low back pain; (3) HIV-positive individuals without pain; and (4) HIV-positive individuals with self-reported CWP. Demographic and clinical information was recorded for all participants, as previously published [1]. The average ages of participants in different groups were between 45 and 54 years and were predominantly African American (64–77%). The proportion of men in the HIV-positive pain groups was higher than in the HIV-negative groups, which reflects the patient population of the HIV clinic at UAB. The average current and nadir CD4^+^ cell counts and the average current and highest viral load (VL) were not different between HIV-positive individuals with or without CWP. Plasma heme levels were significantly higher in PWH with pain (80.9 µM) compared to HIV-negative without pain (14.4 µM), HIV-negative with pain (26.3 µM), and HIV-positive without pain (27.4 µM). All HIV-1-positive individuals recruited in the study were on antiretroviral therapy (Table 1). Our data showed that people with HIV with CWP exhibit significantly higher plasma levels of histamine (Figure 1A) and bradykinin (Figure 1B) compared to HIV-1 negative individuals with or without pain and HIV-1 positive people without pain.

**Demographic and clinical data of HIV-positive and HIV-negative participants.** Age in years; CD4^+^ cells/mm^3^; VL: viral load copies/mL; SD: standard deviation.

3.2 High heme and low HO-1 cause mast cell degranulation in vitro. Previously, we demonstrated that PWH with CWP have increased plasma levels of cell-free heme and attenuated levels of the heme degrading enzyme, HO-1 [1]. In this study, we determined whether high heme and low HO-1 are responsible for mast cell degranulation. We used the RBL-2H3 cell line as it releases histamine, bears IgE receptors, and is an established model for the study of mast cells in vitro. RBL-2H3 cells were treated with DMSO (vehicle) or hemin (25 µM) in the presence or absence of a CO donor (i.e., CORM-A1 (10 mM)) for 2 h. This dose of hemin and CORM-A1 was previously used by us in pain-related studies in vitro in macrophages [1]. The ionophore compound A23187 (5 µg/mL), which induces mast cell degranulation, was used as a positive control. Cells were stained with toluidine blue, as this staining of the acid-fast granules is an established method for the identification and quantification of mast cells [23]. Vehicle-treated RBL-2H3 cells appeared elongated and spindle-shaped and had purple granules stored in cells. However, the shape of the hemin-treated cells was round and irregular, and purple granules were released outside of the cell. Pre-treatment with CORM-A1 markedly inhibited hemin-induced morphological changes and mast cell degranulation (Figure 2A). As a result of cellular degranulation, histamine (Figure 2B) and β-hexosaminidase (Figure 2C) release increased in hemin-challenged cells. However, histamine and β-hexosaminidase release were significantly reduced in CORM-A1-treated cells, suggesting that CO donors can act as mast cell stabilizers.

Next, to determine whether low HO-1 can induce mast cell degranulation, we attenuated HO-1 levels by treating RBL-2H3 cells with HO-1 siRNA. The analysis of HO-1 levels by immunoblot (Figure 2D) showed a significant decline in the protein in HO-1 siRNA-treated cells compared to scrambled siRNA-treated cells. Cells with diminished HO-1 enzyme released higher amounts of histamine (Figure 2E) and β-hexosaminidase (Figure 2F) with or without treatment with hemin compared to the control cells, suggesting that the reduction in HO-1 increases mast cell degranulation. However, adding CORM-A1 to these HO-1-deficient cells significantly reduced histamine and β-hexosaminidase levels.

3.3 Mice lacking HO-1 exhibit increased plasma levels of histamine, bradykinin, and allodynia. To mimic our in vitro data in animals, we determined whether global knockdown of HO-1 in mice was associated with mast cell activation and hypersensitivity. Biochemical analysis of plasma histamine and bradykinin was performed in WT and HO-1^−/−^ mice. We found that HO-1^−/−^ mice had significantly higher plasma levels of histamine (Figure 3A) and bradykinin (Figure 3B) compared to WT mice, suggesting that lower levels of HO-1 in HIV are associated with the release of pain mediators. The staining of skin sections from the neck area with H&E and toluidine blue showed significantly greater numbers of mast cells in the dermal layer (Figure 3C), suggesting that HO-1 depletion activates mast cells. The acetone evaporation test for thermal allodynia showed a higher allodynia score in HO-1^−/−^ mice compared to their WT counterparts (Figure 3D). However, the hot plate test for thermal hyperalgesia revealed no change in latency to response between these groups of animals (Figure 3E). We have previously shown that HO-1^−/−^ mice have a decreased threshold to mechanical stimulation [1], and therefore, mechanical allodynia was not tested in this study.

3.4 CO donors attenuated heme-induced mast cell activation and allodynia in a mouse model of hemolysis. To further mimic our in vitro studies in animals, we determined whether hemolytic mice exhibit mast cell activation. Hemolysis was induced in adult male and female C57BL/6 mice by giving a bolus dose of PHZ on two consecutive days. Animals were then treated with either the heme-scavenging protein, hemopexin, or the carbon monoxide donor, CORM-A1, one hour after the second PHZ injection. The CORM-A1 injection was repeated for the next three consecutive days, and animals were harvested on day 4 after the second PHZ injection (Figure 4A). Plasma levels of cell-free heme were elevated in PHZ-exposed mice (Figure 4B). However, hemopexin, but not CORM-A1, treatment reduced plasma heme levels in PHZ-exposed mice. H&E and toluidine blue staining of neck skin sections (Figure 4C) demonstrated that hemopexin and CORM-A1 treatment significantly reduced the PHZ-dependent increase in mast cells in the dermal layer (Figure 4D). Plasma concentrations of histamine (Figure 4E) and bradykinin (Figure 4F) were elevated in PHZ-exposed animals. In contrast, these factors were significantly lower in mice that received either hemopexin or CORM-A1 along with PHZ. Finally, the analysis of evoked pain behaviors showed that PHZ-induced mechanical (Figure 4G) and thermal (cold) allodynia (Figure 4H) are attenuated in CORM-A1-treated mice. However, PHZ-dependent thermal (heat) hyperalgesia was not altered with CORM-A1 administration (Figure 4I). Together, these studies show that CO donors mitigate mast cell activation/degranulation and the associated mechanical and cold allodynia.

## 4. Discussion

Mast cells are tissue-resident immune cells that release immunomodulators, including pro- and anti-inflammatory cytokines, chemoattractants, vasoactive compounds, neuropeptides, and growth factors, in response to allergens and pathogens, constituting a first line of host defense [24,25,26]. Mast cells can be activated by IgE through its receptor, FceRI, but also by toll-like receptors or interleukin-1 (IL-1) [27,28]. They can be activated during several viral infections, such as cytomegalovirus and dengue virus [29,30]. During HIV-1 infection, there are conflicting reports that mast cells may act as reservoirs of latent infection [31] and also promote viral trans-infection of CD4^+^ T cells [32]. However, whether mast cells are activated in HIV-1 infection is unclear. In HIV-1-positive individuals on active antiretroviral therapy with an undetectable viral load, it is very unlikely that the virus plays any significant role in mast cell activation. Nevertheless, our data shows that HIV-1-positive individuals with chronic pain have elevated plasma levels of histamine and bradykinin, suggesting that mast cells are activated in this group. These cohorts also have elevated plasma levels of cytokines like IL-1β, IL-6, and TNF-α [1], which can be released by mast cell activation [33,34,35,36]. Therefore, identifying factors that activate mast cells in PWH with chronic pain may be crucial to pain management strategies in this population.

The current evidence suggests that PWH have increased levels of reactive oxygen species, which increase hemolysis and cell-free heme levels in blood [1,37,38,39,40]. In our previous study, we found that hemolysis is particularly high in PWH who have chronic pain [1]. These individuals also have subdued levels of the heme-scavenging protein, hemopexin, and the heme-metabolizing enzyme, HO-1 [1]. Under physiological conditions, cell-free heme is maintained in plasma at low levels by hemopexin [40,41]. Cell-free heme binds with hemopexin, and the complex is internalized by the CD91 receptor [42]. HO-1 constitutes one of the three isozymes of HO that catalyzes the degradation of heme into biliverdin and CO, which exert potent anti-oxidative stress and anti-inflammatory functions [43,44,45]. HO-1 exerts its protective role in neurodegenerative, cardiovascular, metabolic, and several inflammatory diseases [46]. HO-1-deficient mice are more sensitive to heme-induced oxidative stress [47]. HO-1 has been shown to promote the stabilization of mast cells [19] and dampen cytokine production by activated mast cells [48]. Our in vitro study also found that genetically silencing HO-1 in mast cells (RBL-2H3) by siRNA or exposing mast cells to hemin increased activation and degranulation. This was evidenced by the enhanced release of histamine and β-hexosaminidase and the rounding of spindle-shaped cells on toluidine blue stain. Moreover, our in vivo studies in animals validated the finding seen in cells. There were significantly elevated levels of plasma histamine and bradykinin, along with an increased mast cell count in the dermal layer of the skin of HO-1^−/−^ mice and in C57BL/6 mice exposed to the hemolytic agent PHZ. Therefore, it is evident from our cell- and animal-based studies that high cell-free heme and low HO-1 are pathognomonic of mast cell activation. Therefore, these two factors may be playing an important role, at least in part, in HIV-dependent increases in plasma histamine and bradykinin levels and the associated chronic pain.

Although it is not entirely clear how cell-free heme or low HO-1 causes mast cell activation/degranulation, there is some evidence that the resulting increase in reactive species from these two pathological events may be responsible [49]. However, inhibition of reactive species production has had little effect on the secretion of cytokines and degranulation of mast cells in the past [50,51]. In this study, we explored whether therapeutical administration of a CO donor, which is a product of heme metabolism by HO-1, dampens heme-induced mast cell activation. CO has important signaling, antiapoptotic, and anti-inflammatory effects [52]. CO-releasing molecules (CORMs) represent a group of compounds capable of carrying and liberating controlled and sustained quantities of CO to tissues and organs and have been shown to have anti-inflammatory effects [53] and reduce reactive oxygen species production and oxidative stress [54,55,56]. In our study, we used CORM-A1, which releases 1 molecule of CO per mole of CORM-A1. The biological half-life of CORM-A1 at 37 °C and pH 7.4 is approximately 27 minutes [57]. Treatment with CORM-A1 reduced hemin-induced histamine and β-hexosaminidase release from RB-2H3 cells. CORM-A1 administration also abrogated plasma histamine and bradykinin levels in hemolytic mice. These effects of CORM-A1 were mimicked by administering hemopexin to hemolytic mice. Therefore, these studies clearly depict that removing the source of oxidative stress, such as cell-free heme by hemopexin or using CORM-A1, which slowly releases CO to the environment and closely reflects the action of natural HO-1 [57], may be a more superior strategy to mitigate mast cell activation compared to using traditional antioxidants.

Our previous and current evoked pain studies in rodents have shown that an increase in cell-free heme and genetic deletion of HO-1 induce mechanical and cold allodynia [1]. While hemolysis also induced hyperalgesia due to heat sensation, HO-1 deletion did not affect heat sensation in animals. Therefore, it is possible that heme is able to modulate the following receptors that are involved in mechanical, cold, and heat sensations (mechanical: transient receptor potential vanilloid 4 (TRPV4) and transient receptor potential cation channel, subfamily A (TRPA1); cold: transient receptor potential cation channel, subfamily M member 8 (TRPM8); heat: TRPV1). However, HO-1 may only be able to modulate mechanical and cold receptors. We have previously shown that treating hemolytic mice with hemopexin attenuates heme-induced allodynia [1]. Therefore, in this study, we tested whether administering CORM-A1 would do the same in hemolytic mice. Interestingly, CORM-A1 administration only mitigated mechanical and cold allodynia but not hyperalgesia to heat, suggesting that CORM-A1 mimics the effects of HO-1. In our future studies, we will test how heme and HO-1 modulate different pain receptors and whether heme scavenging and CORM-A1 may reverse those changes.

Mast cells are recognized for their role in allergy and anaphylaxis, but increasing evidence supports their role in neurogenic inflammation leading to pain and itch [11,58,59]. Few activated mast cells are capable of inducing neuro-inflammation and increasing nociception [8]. The release of neuro-inflammatory mediators such as histamine and bradykinin upon mast cell activation/degranulation generates action potentials in sensory neurons and promotes the release of neurotransmitters and/or neuropeptides associated with pain [11,24]. Histamine contributes to the generation of pain hypersensitivity through the sensitization of polymodal nociceptors, resulting in increased firing rates [60]. Bradykinin enhances the excitability of nociceptor sensory neurons and also increases their sensitivity, which exacerbates pain [61]. The release of neuropeptides and/or neurotransmitters such as substance P from sensory neurons can further activate mast cells, creating a loop of neuro-inflammation and pain [11]. Therefore, breaking this loop would greatly contribute to ameliorating chronic pain and neuroinflammation in HIV.

## 5. Conclusions

In conclusion, our present study successfully elucidates that heme-dependent mast cell degranulation and the release of pain mediators may contribute to CWP in HIV; therefore, targeting mast cells may reduce algesia in this cohort. Therapeutic strategies such as heme scavenging and CO donors may also find usefulness in other hemolytic disorders associated with pain, such as sickle cell disease, in which plasma levels of histamine and bradykinin are reported to be high [62,63].

## Figures and Tables

**Figure 1 antioxidants-12-01213-f001:**
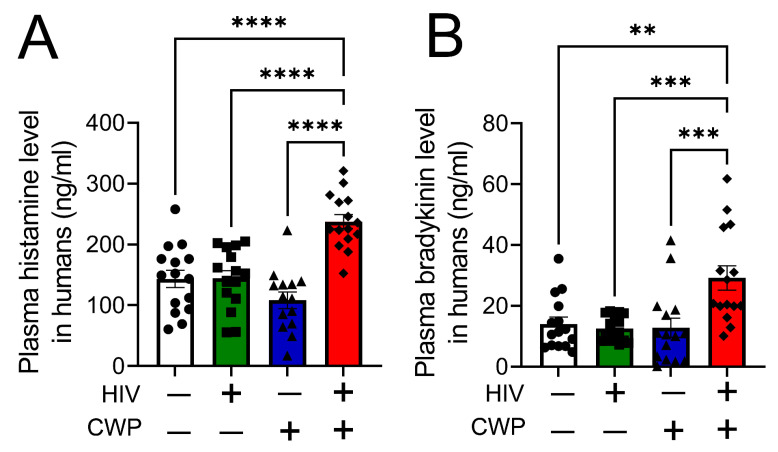
**People with HIV who have chronic widespread pain (CWP) have increased plasma levels of histamine and bradykinin.** HIV-1 positive individuals with CWP exhibit significantly higher plasma levels of histamine (n = 14–15) (**A**) and bradykinin (n = 15) (**B**) compared to HIV-1 negative individuals with or without CWP and HIV-1 positive people without pain. Individual values and means ± SEM. ** *p* < 0.01, *** *p* < 0.001, **** *p* < 0.0001 vs. groups at the end of individual lines; one-way ANOVA followed by Tukey’s post-hoc testing.

**Figure 2 antioxidants-12-01213-f002:**
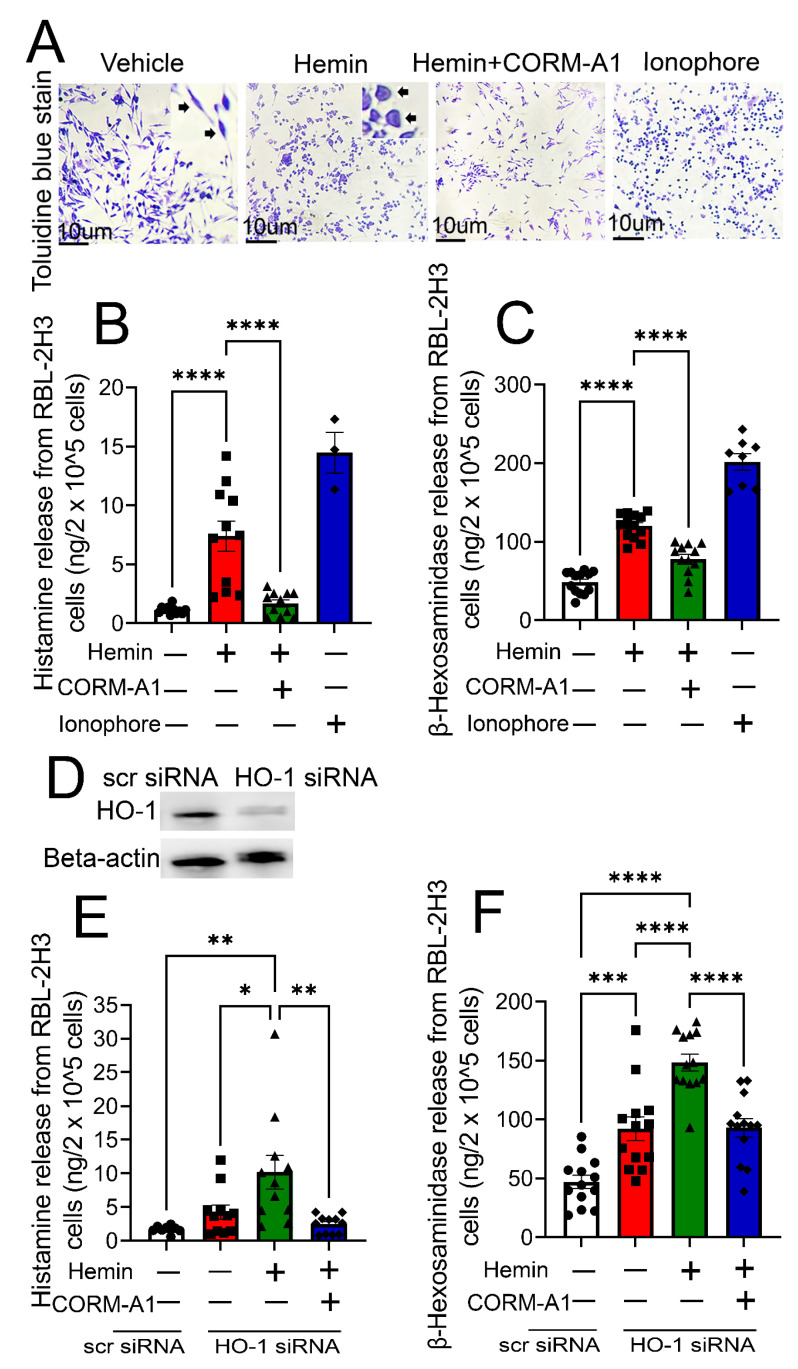
**High heme and low HO-1 cause mast cell degranulation in vitro.** RBL-2H3 cells were treated with DMSO (vehicle) or hemin (25 µM) in the presence or absence of a CO donor (i.e., CORM-A1 (10 µM)) for 2 h. Upon toluidine blue staining, vehicle-treated cells were elongated and spindle-shaped (shown by arrows in the magnified image), while hemin-treated cells were round and irregular (shown by arrows in the magnified image), and purple granules were released outside of the cell. CORM-A1 reversed these changes. A23187 (5 µg/mL) was used as a positive control (**A**). Hemin challenge increased histamine (n = 11) (**B**) and β-hexosaminidase (n = 13) (**C**) release in vehicle treated cells but not in CORM-A1-treated cells. RBL-2H3 cells grown to 60% confluency were treated with either control (scrambled, scr) or HO-1 siRNA in vitro. HO-1siRNA reduced HO-1 levels in these cells (n = 5) (**D**). HO-1 depletion in RBL-2H3 cells released higher amounts of histamine (n = 8–11) (**E**) and β-hexosaminidase (n = 13) (**F**) with or without treatment with hemin compared to the control cells. Treatment of HO-1 deficient cells with CORM-A1 (10 mM) significantly reduced hemin-induced histamine (**E**) and β-hexosaminidase (**F**) release compared to vehicle (DMSO)-treated cells. Individual values and means ± SEM. * *p* < 0.05, ** *p* < 0.01, *** *p* < 0.001, **** *p* < 0.0001 vs. groups at the end of individual lines; one-way ANOVA followed by Tukey’s post-hoc testing.

**Figure 3 antioxidants-12-01213-f003:**
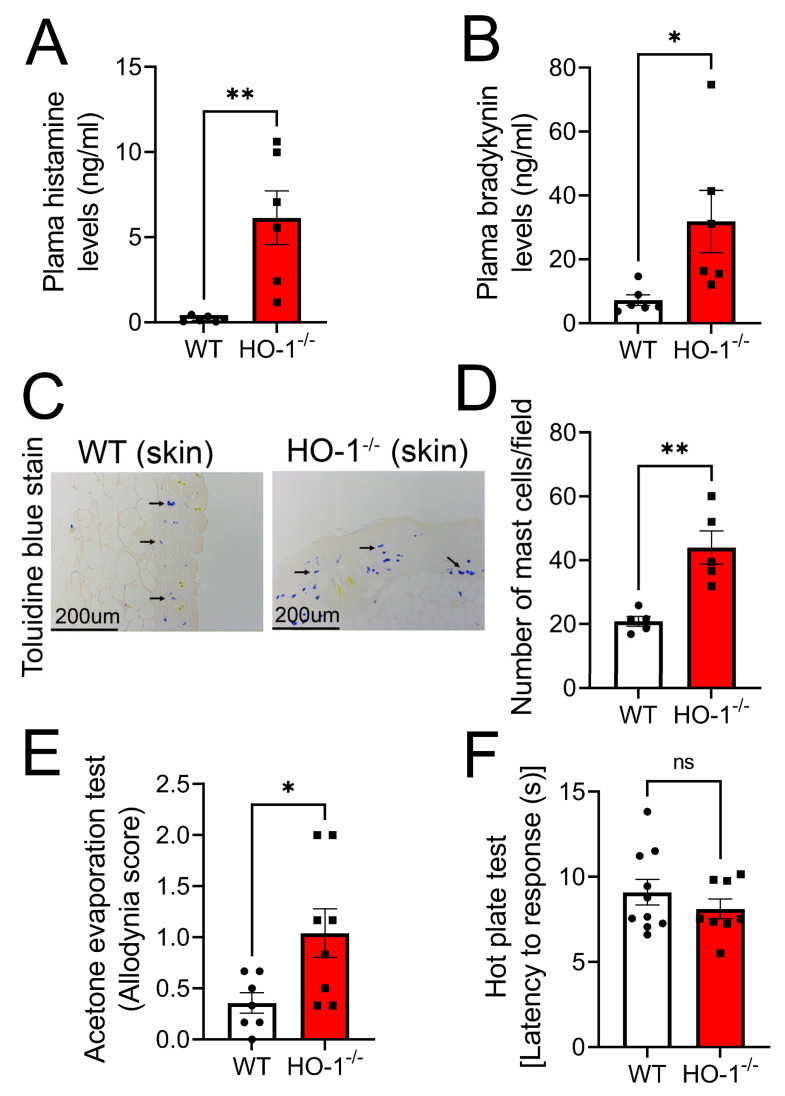
**Mice lacking HO-1 exhibit increased plasma levels of histamine and bradykinin and allodynia.** HO-1^−/−^ mice had significantly higher plasma levels of histamine (n = 6) (**A**) and bradykinin (n = 6) (**B**) compared to wildtype (WT) mice. H&E and toluidine blue staining of skin from the neck (n = 5) (**C**) showed a significantly higher number of mast cells in the dermal layer of HO-1^−/−^ compared to WT mice (n = 5) (**D**). The acetone evaporation test showed a higher thermal allodynia score in HO-1^−/−^ mice compared to their WT counterparts (n = 7–8) (**E**). The hot plate test showed no significant change in thermal hyperalgesia between the two groups (n = 8–10) (**F**). Individual values and means ± SEM. * *p* < 0.05, ** *p* < 0.01 vs. groups at the end of individual lines; one-way ANOVA followed by Tukey’s post-hoc testing.

**Figure 4 antioxidants-12-01213-f004:**
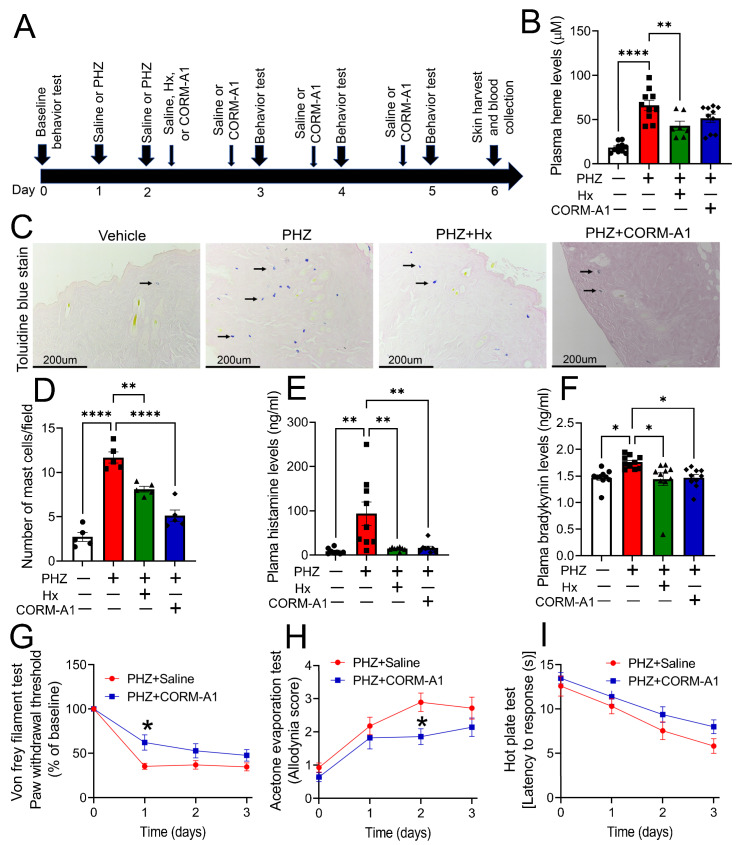
**A CO donor attenuates heme-induced mast cell activation and allodynia in a mouse model of hemolysis.** Phenylhydrazine hydrochloride (PHZ, 50 mg/kg, IP) or saline was administered to adult male and female C57BL/6 mice on two consecutive days. One hour following the second PHZ injection, a subset of mice received saline (vehicle), purified human hemopexin (Hx) (4 mg/kg, IP), or CORM-A1 (2.0 mg/kg, IP). The CORM-A1 injection was repeated for the next three consecutive days, and animals were harvested on day 4 after the second PHZ injection (**A**). Hx, but not CORM-A1, reduced the PHZ-mediated increase in plasma levels of cell-free heme (n = 7–10) (**B**). H&E and toluidine blue staining of neck skin sections (n = 5) (**C**) demonstrated that Hx and CORM-A1 significantly reduced PHZ-dependent increase in mast cells in the dermal layer (n = 5) (**D**). Hx and CORM-A1 also attenuated the PHZ-induced increase in plasma levels of histamine (n = 8–10) (**E**) and bradykinin (n = 10) (**F**). Evoked pain behaviors showed that CORM-A1 dampened PHZ-induced mechanical (n = 14–16) (**G**) and thermal (cold) allodynia (n = 14) (**H**), but not thermal (heat) hyperalgesia (n = 11–13) (**I**). Individual values and means ± SEM. * *p* < 0.05, ** *p* < 0.01, **** *p* < 0.0001 vs. groups at the end of individual lines; one-way ANOVA followed by Tukey’s post-hoc testing.

**Table 1 antioxidants-12-01213-t001:** **Demographic and clinical data of study participants:** The demographic and clinical data is shown, which describes average age in years, current and nadir CD4^+^ cell count (cells/mm^3^), average current and highest viral load (VL), and mean plasma heme levels (µM). SD: standard deviation. ^a^
*p*  <  0.05 vs. HIV-negative without pain; ^b^
*p*  <  0.05 vs. HIV-negative with pain; ^c^
*p*  <  0.05 vs. HIV-positive without pain; one-way ANOVA followed by Tukey post-hoc testing.

Group (Number of Participants)	HIV_neg_, Pain_neg_ (14)	HIV_neg_, Pain_pos_ (14)	HIV_pos_, Pain_neg_ (14)	HIV_pos_, Pain_pos_ (13)
Avg. age (SD)	50.1 (12.6)	45.3 (15.0)	48.6 (8.4)	54.1 (5.8)
Percent Females	64.3	64.3	50	38.5
Percent Afr. Amer.	64.3	64.3	71.4	76.9
Avg. CD4^+^ (SD)			715.4 (421.2)	637.5 (385.0)
Avg. Nadir CD4^+^ (SD)			165.0 (231.5)	204.5 (202.4)
Avg. VL (SD)			19.5 (1.9)	59.2 (130.7)
Avg. Highest VL (SD)			199,115 (249,762)	428,995 (1,220,712)
Plasma heme (µM) (SD)	14.4 (4.0)	26.3 (13.0)	27.4 (12.4)	80.9 (62.4) ^a,b,c^

## Data Availability

The data presented in this study are available on request from the corresponding author.

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
