# Peer review of "High Heme and Low Heme Oxygenase-1 Are Associated with Mast Cell Activation/Degranulation in HIV-Induced Chronic Widespread Pain"

_antioxidants, 2023, doi:10.3390/antiox12061213_

Round 1

Reviewer 1 Report

This is an interesting paper reflecting carefully performed studies.

While the actual studies reported are clear and appropriately described, so modifications are required in aspects of presentation.

The abstract and the concluding paragraph of the Introduction need to be modified to indicate that the authors are describing and assessment of histamine/bradykinin and free heme and free heme in an appropriately controlled patient populations but that the other, correlative, studies are performed in cell models or in vivo mouse models. The abstract is not clear on this, and the last paragraph of the introduction should state it explicitly.

2. In the Methods, the authors should state how human heme levels (Table 1) were measured. Was it done by the same technique as for mice, or was it done in a different methodology by the clinical lab?

3. Also in the Methods, the Von Frey testing should be more correctly described as the Von Frey microfilament test and not just "Von Frey".

4. In the Discussion, the paragraph on line 411 should be moved up to precede the paragraph beginning "Mast cells are recognized…" This provides a better flow of the concepts.

Reviewer 2 Report

Manuscript entitled “ High heme and low heme oxygenase-1 is associated with mast  cell activation/degranulation in HIV-induced chronic widespread pain” by Chatterjee et al investigated the underlying mechanism of chronic pain in HIV patients. Authors previously demonstrated high heme and low  heme oxygenase 1 (HO-1) levels in the patients. Here they have demonstrated that HIV patients with chronic pain have high plasma level of histamine and bradykinin. Heme cause mast cell degranulation to cause release of histamine in RBL-2H3 cells which is inhibited by CO donor CORM-A1. Moreover, HO-1 knockout mice demonstrated enhanced plasma histamine and bradykinin. CORM-A1 treatment reduced mast cell activation and plasma histamine level in mice.

This study is well conceived, and manuscript is nicely written. This study first time demonstrated the involvement of mast cell mediators in chronic pain in HIV patients. However there are some concerns that is needed to be addressed.

1) In Fig. 4, although CORM-A1 doesn’t cause significant change in plasma heme, but still, it can reduce MC activation and histamine release?

2) Fig 2a, mast cells morphological change is not clearly visible in that magnification. Authors should give high magnification image.

Round 2

Reviewer 2 Report

Authors addressed all my concerns. Revised manuscript looks much better.